# HIV testing behaviors and willingness to receive oral rapid HIV testing among dental patients in Xi'an, China

**Bei Gao[1], Lirong Wang[2]\*, Anthony J. Santella[3], Guihua Zhuang[2], Ruizhe Huang[4], Boya Xu[5], Yujiao Liu[2], Shuya Xiao[2], Shifan Wang[2]**

1 Department of Orthodontics, Stomatology Hospital, Xi'an Jiaotong University Health Science Center, Xi'an, Shaanxi, China, 2 Department of Epidemiology and Biostatistics, School of Public Health, Xi'an Jiaotong University Health Science Center, Xi'an, Shaanxi, China, 3 Department of Health Professions, School of Health Professions and Human Services, Hofstra University, Hempstead, New York, United States of America, 4 Department of Preventive Dentistry, Stomatology Hospital, Xi'an Jiaotong University Health Science Center, Xi'an, Shaanxi, China, 5 Department of Oral Implantology, Stomatology Hospital, Xi'an Jiaotong University Health Science Center, Xi'an, Shaanxi, China

\* wanglr@mail.xjtu.edu.cn

**Data Availability Statement:** All relevant data are within the manuscript and its Supporting information files.

## Abstract

### Introduction

HIV testing is an important strategy for controlling and ultimately ending the global pandemic. Oral rapid HIV testing (ORHT) is an evidence-based strategy and the evidence-based shows is favored over traditional blood tests in many key populations. The dental setting has been found to be a trusted, convenient, and yet untapped venue to conduct ORHT. This study assessed the HIV testing behaviors and willingness to receive ORHT among dental patients in Xi'an, China.

### Methods

A cross-sectional survey of dental patients from Xi'an was conducted from August to September 2017. Dental patients were recruited using a stratified cluster sampling. A 44-item survey was used to measure HIV/AIDS knowledge, HIV testing behaviors, and willingness to receive ORHT.

### Results

Nine hundred and nine dental patients completed the survey with a mean HIV/AIDS knowledge score of 10.7/15 (SD 2.8). Eighty-four participants (9.2%) had previously received an HIV test. Participants would have a high rate of HIV testing if they had higher monthly income ($OR = 1.982$, 95% $CI$: 1.251–3.140) and a higher HIV/AIDS knowledge score ($OR = 1.137$, 95% $CI$: 1.032–1.252). Five hundred and eighty-two participants (64.0%) were willing to receive ORHT before a dental treatment, 198 (21.8%) were not sure, and 129 (14.2%) were unwilling. Logistic regression showed that age ($OR = 0.970$, 95% $CI$: 0.959–0.982), HIV/AIDS knowledge score ($OR = 1.087$, 95% $CI$: 1.031–1.145), previous HIV test ($OR = 2.057$, 95% $CI$: 1.136–3.723), having advanced HIV testing knowledge ($OR = 1.570$, 95%

**Funding:** We would like to thank the National T&S Major Project of China (grant no. 2018ZX10721202) for supporting us. The sponsors were not involved in the study design; collection, analysis and interpretation of data; in the writing of the manuscript; or in the decision to submit the manuscript for publication.

**Competing interests:** The authors have declared that no competing interests exist.

*CI*: 1.158–2.128), and having advanced ORHT knowledge (*OR* = 2.074, 95%: *CI* 1.469–2.928) were the factors affecting the willingness to receive ORHT.

## Conclusions

The majority of dental patients had not previously received an HIV test, although many were receptive to being tested in the dental setting. The dental setting as a venue to screen people for HIV needs further exploration, particularly because many people do not associate dentistry with chairside screenings. Increasing awareness of ORHT and reducing testing price can further improve the patient's willingness to receive ORHT.

## Introduction

Although the life expectancy of people living with HIV (PLWH) has improved in the past decades [1], HIV/AIDS remains an issue of public health significance. By the end of 2019, 32.7 million people have died from AIDS-related illnesses since the start of the epidemic and 38.0 million people globally were living with HIV [2].

HIV testing is a vital strategy for controlling the pandemic. Early detection of HIV infection is an effective way to decrease HIV transmission. People aware of their infection have the opportunity to reduce their risk behaviors and receive antiretroviral treatment (ART) to achieve viral suppression [3]. It is reported that the transmission rate among persons unaware was three to seven times than those aware of their infections [4].

Unfortunately, HIV infections are often undiagnosed and diagnosed late including concurrently with AIDS [5]. In 2019, only 81% of PLWH knew their status worldwide [2]. In China, cumulative 850,000 PLWH were reported by the end of 2018, based on the estimates of 1,250,000, almost one-third of the PLWH have not yet been tested and diagnosed [6]. Zhang [7] estimated that the average time from infection and being tested was 6.8 ± 2.7 years in China. In Xi'an, a populous urban city in Northwest China, the late diagnosis rate of the newly identified HIV/AIDS was 31.1% (436/1,402) in 2017 [8].

Lacking testing venues is an important reason for low and delayed diagnoses. To expand testing venues, opt-out HIV testing has been implemented in medical settings since 2008 among priority populations and settings such as perinatal exams, prenatal exams, pre-operative exams, sexually transmitted disease exams and other types of testing services in China [9]. Almost half (44.7%) of 528,234 persons diagnosed with HIV between 2006 and 2014 in China were diagnosed at hospitals, indicating that most of the HIV testing in China was conducted in health and hospital systems [10]. However, dental patients are nor high-risk groups or sentinel surveillance groups. For both opt-out and opt-in testing, it is necessary to first understand the population's HIV testing willingness. Otherwise, even if opt-out is carried out, a considerable number of people may refuse to be tested. The Chinese government has also explored the feasibility of offering HIV testing in at primary healthcare settings such as community health centers [11]. However, since China is a low-prevalence country, such a practice is still being explored. More research is needed to evaluate the effectiveness of this setting and other nontraditional settings such as dentistry.

Oral manifestations are among the earliest and most important clinical indicators in HIV infection and are important facilitators for recognizing, monitoring, and predicting the course of the disease [12, 13]. Few people with HIV infection fail to experience oral lesions during the

course of their disease [14]. According to previous studies, rampant carries, severe periodontitis and oral candidiasis are the most notable oral lesions in PLWH [13, 15, 16].

The first oral rapid HIV test (ORHT) was approved by the United States (U.S.) Food and Drug Administration (FDA) in 2004 [17]. In the U.S. and other high-income countries, scholars have proposed that ORHT has the potential to improve the diagnosis rate of PLWH [18, 19]. The tests have high sensitivity and specificity; both above 99% [20–22]. It also offers many advantages over blood-based testing and could address several challenges in implementing HIV testing in settings without laboratories [20]. This testing method does not require blood collection, has no risk of occupational exposure and cross-infection, and can prevent the patient from refusing to test because of fear of blood draw [23].

Studies about HIV testing in dental settings showed that most dentists have positive attitudes to ORHT and are willing to administer the test [24–27]. A cross-sectional survey of dentists from Xi'an revealed that 91.2% of 477 respondents thought that ORHT was needed in private dental practice [24], 88.6% of 475 dentists belonging to the Korean Dental Association thought ORHT was necessary in dental settings [25], and a study from four cities in India reported that the proportion was 79.9% (402/503) [26], 65.1% of 532 responders from a survey conducted in Australia believed that ORHT was needed in private dental practice [27]. Dentists have the skill set to deliver difficult news and to ensure appropriate treatment or referral [28, 29]. According to a representative survey of dentists from the U.S., 56.7% (1022/1802) were willing to offer HIV testing [30], and in another survey conducted in Vietnam, the proportion was 90% (38/42) [31].

One of the concerns about rapid HIV screening is the dental patients' acceptance [32]. Research about rapid HIV testing acceptance has been conducted in high HIV prevalence jurisdictions. A majority, 72.2% (293/406), of dental patients in New York City (U.S.) were agreed to have HIV testing performed in dental hygiene clinics [33]. Additionally, 84.5% of the 600 patients surveyed in South Florida (U.S.) accepted the ORHT offered by a dentist [34]. These studies focused on cities with high HIV prevalence; whether these attitudes would hold true in moderate and low prevalent cities and suburban and rural areas, remains unknown.

Given the lack of studies about dental patients' acceptance towards ORHT in China, we conducted a cross-sectional survey Xi'an, located in Northwest China. The number of HIV/AIDS cases in Xi'an has increased from year to year from 2005 to 2017 [35]. The aim of the present study was to explore HIV testing behaviors and willingness to receive an ORHT among dental patients in Xi'an.

## Methods

### Study design and participants

A cross-sectional survey was conducted among dental patients in Xi'an from August to September 2017. First, practices from a sampling frame of dental practices in Xi'an were divided into three groups according to their setting type (dental hospital, department of dentistry in a general hospital and private dental practice). Then, practices were selected randomly amongst all three types. Dental patients 18 years and older who visited the sample practices from August to September were invited to participate in the survey. The proportion of participants who were willing to receive ORHT ranged from 24% to 91% [36]. If we take 24% as the expected proportion of our survey, $\alpha = 0.01$ and the allowable error d = 0.15P, according to the formula $n = \frac{t_\alpha^2 PQ}{d^2}$, which is commonly used to estimate the sample size of cross sectional study, then the approximately sample size would be 934.

The study team presented a short introduction about ORHT to all the participants (enough so they would be aware of what the survey was about but not too much that it would influence

their results) and after obtaining their consent surveyed their knowledge and attitudes toward HIV/AIDS and ORHT. The investigators were trained graduate students. One member of the team was responsible for implementing the survey, the other was responsible for explaining the purpose of the survey, consenting the participant, and checking the survey after it was completed to ensure it was complete.

All study procedures and consent processes were approved by the ethics committee (Institutional Review Board) of Xi'an Jiaotong University. The written informed consent was obtained from each participant.

## Data collection

The survey instrument was modified from a validated questionnaire from researchers in the U.S. [33]. An expert consensus panel of Chinese dental, sexual health, and epidemiology scholars reviewed a draft questionnaire to conform the validity. The questionnaire was pre-tested among 20 patients in Xi'an, and we modified the questionnaire based on the results, including the order of the questions, some inappropriate options and inaccurate expression of the questions. The final questionnaire consisted of 44 questions, assessing socio-demographic characteristics (eight items), HIV/AIDS knowledge (modified HIV-KQ-18 [37] (fifteen items), HIV testing behaviors (seven items), HIV testing knowledge (six items), ORHT knowledge (three items) and the willingness to receive ORHT (five items). We assessed ORHT knowledge by asking participants what the advantages of ORHT were, the options were high accuracy, quick results, and no need to draw blood. To collect the willingness to receive ORHT, we asked the participants: "If a dentist can carry out Oral Rapid HIV Testing, would you be willing to test it before treating oral diseases?". The answers were yes, unwilling and unsure. For participants who chose "yes", we categorized them as willing to use ORHT.

## Analysis

Statistical Package for the Social Sciences (SPSS), version 13.0 was used for statistical analyses. Questions about HIV knowledge contained 15 items and each correct item was awarded one mark and the maximum score was 15. Knowledge of HIV testing and ORHT contained six items and three items, respectively. Responders who answered one or more questions correctly were identified as having advanced HIV testing or ORHT knowledge.

Differences between groups were tested using t-test or one-way ANOVA for continuous variables and chi-squared statistic for categorical variables. Unconditional logistic regression was used to analyze factors influencing HIV testing behaviors and willingness to receive an ORHT. Statistical significance was assessed using a two-sided test at the $\alpha = 0.05$ level for all studies.

## Results

### Participant characteristics

Of the 909 participants, the mean age was 34.4 years (SD 12.3), 398 (43.8%) were male, 482 (53.1%) had completed college and/or graduate school, 551 (60.6%) were married, 286 (32.1%) had a monthly income more than 5000 yuan RMB (708.4 USD); 506 (55.7%) were from dental hospitals, 307 (33.8%) were from department of dentistry in general hospitals, 96 (10.6%) were from private dental practice. The average HIV knowledge score was 10.7 (out of 15) (SD = 2.8). The three questions with the lowest correct answer rate were: mosquito biting can't transmit HIV/AIDS (24.4%), coughing and sneezing can't transmit HIV/AIDS (55.7%), and

**Table 1. Participants' characteristics and HIV testing behavior.**

| | Total (N = 909) | Received an HIV Test previously | | $t/\chi^2$ | P |
| --- | --- | --- | --- | --- | --- |
| | | Yes (n = 84) | No (n = 825) | | |
| Age, X̄ (SD) | 34.4(12.3) | 36.1(11.4) | 34.2(12.4) | -1.366 | 0.172 |
| Sex, n (%) | | | | | |
| Male | 398(43.8) | 34 (8.5) | 364 (91.5) | 0.412 | 0.521 |
| Female | 511(56.2) | 50 (9.8) | 461 (90.2) | | |
| Nationality n (%) | | | | | |
| Han | 889(97.8) | 80 (9.0) | 809 (91.1) | 2.823 | 0.093 |
| Others | 20(2.2) | 4(20.0) | 16 (80.0) | | |
| Education*, n (%) | | | | | |
| High school and lower | 425(46.9) | 28 (6.6) | 397 (93.4) | 6.800 | 0.009 |
| College and above | 482(53.1) | 56 (11.6) | 426 (88.4) | | |
| Marriage*, n (%) | | | | | |
| Unmarried | 357(39.3) | 23(6.4) | 334 (93.6) | 5.489 | 0.019 |
| Married | 551(60.6) | 61 (11.1) | 491 (88.9) | | |
| Monthly income*, n (%) | | | | | |
| ≥ 5000 yuan RMB (708.4 USD) | 286(32.1) | 40 (14.0) | 246 (86.0) | 10.931 | 0.001 |
| <5000 yuan RMB (708.4 USD) | 606(67.9) | 43 (7.1) | 563 (92.9) | | |
| Employment Status, n (%) | | | | | |
| Employed | 559(61.5) | 53 (9.5) | 506 (90.5) | 0.100 | 0.752 |
| Unemployed | 345(38.0) | 31(8.9) | 319 (91.1) | | |
| Type of dental setting visited, n (%) | | | | | |
| Dental hospital | 506(55.7) | 43(8.5) | 463(91.5) | 1.999 | 0.368 |
| Department of dentistry | 307(33.8) | 34(11.1) | 273(88.9) | | |
| Private dental practice | 96(10.6) | 7(7.3) | 89(92.7) | | |
| HIV Knowledge Score, X̄ (SD) | 10.7(2.8) | 11.6(2.5) | 10.6(2.8) | 2.928 | 0.003 |

* Not all the participants answered the question.

there is no effective vaccine for HIV/AIDS (58.0%). Other participant characteristics are presented in Table 1.

## HIV testing behaviors

Eighty-four participants (9.2%) had previously been tested for HIV, among which 73 (86.9%) were tested in hospitals, 4(4.8%) were in Center for Disease Control and Prevention (CDC) and 5 (6.0%) were tested when donating blood. Among the 39 participants who remembered the price of the test, 16 (41.0%) paid more than 100 yuan RMB (14.3 US dollars), 2 (5.1%) paid 51 to 100 yuan RMB (7.3–14.3 US dollars), 5(12.8%) paid 1 to 50 yuan RMB (0.1–7.2 US dollars), 16 (41.0%) were tested for free.

HIV testing behaviors by different demographic characteristics are shown in Table 1. The HIV testing rate among participants who had completed college and/or graduate school was 11.6%, which is higher than that among participants who had their highest level of education as high school (11.6% vs. 6.6%; $\chi^2 = 6.800$, $p<0.05$). The married participants' HIV testing rate was higher than unmarried (11.1% vs. 6.4%; $\chi^2 = 5.489$, $p<0.05$). The participants whose monthly income was higher than 5000 yuan RMB (708.4 USD) had a higher HIV testing rate than those who had a lower monthly income (14.0% vs. 7.1%; $\chi^2 = 10.931$, $p<0.05$).

**Table 2. Predictors of conducting HIV testing.**

| | B | S.E | Wals | df | Sig. | OR | 95% C.I. | |
|---|---|---|---|---|---|---|---|---|
| | | | | | | | Lower | Upper |
| Monthly Income | 0.684 | 0.235 | 8.480 | 1 | 0.004 | 1.982 | 1.251 | 3.140 |
| HIV Knowledge Score | 0.128 | 0.049 | 6.750 | 1 | 0.009 | 1.137 | 1.032 | 1.252 |
| Constant | -4.652 | 0.644 | 52.238 | 1 | <0.001 | 0.010 | | |

In multiple logistic regression analyses, participants would have a higher rate of HIV testing if they had higher monthly income (*OR* = 1.982, 95% *CI*: 1.251–3.140) or higher HIV knowledge score (*OR* = 1.137, 95% *CI*: 1.032–1.252) (Table 2).

## Willingness to receive an ORHT

After a short introduction about ORHT, participants who had high/advanced HIV testing or ORHT knowledge were 452 (49.7%) and 720 (79.2%), respectively. In details, for the advantages of ORHT, the rate of choosing "quick results", "no need to draw blood" and "high accuracy" are 561(61.7%), 450(49.5%), and 360(39.6%), respectively. And other questions are presented in Table 3. Of the 909 participants, 582 (64.0%) were willing to receive ORHT before a dental treatment, 198(21.78%) were not sure, and 129 (14.2%) were unwilling to receive an ORHT. When asked about the highest price they would pay for an ORHT, 132 (22.8%) chose free, 43 (7.4%) chose 10 yuan RMB (1.4 USD), 119 (20.5%) chose 30 yuan RMB (4.3 USD), 180 (31.0%) chose 50 yuan RMB (7.2 USD), 106(18.3%) chose 80 yuan RMB (11.3 USD), respectively.

Willingness to receive an ORHT by different demographic characteristics are shown in Table 4. Participants who were willing to receive ORHT were younger than the participants who were not willing (32.9 vs. 40.5; *F* = 20.802, *p*<0.05). Additionally, 66.9% of female participants were willing to receive an ORHT, which was higher than for males (66.9% vs. 60.3%; $\chi^2$ = 7.060, *p*<0.05). The mean HIV knowledge score of participants who were willing to receive an ORHT was 11.1 (2.5) out of 15, which was higher than that among participants who were not willing (11.1 vs. 10.2; *F* = 93.006, *p*<0.05). For participants who previously received an HIV test, the proportion willing to receive an ORHT, not willing and not sure was 68 (81.0%), 6 (7.1%) and 10 (11.9%) respectively, which is different from those who had not previously received an HIV test ($\chi^2$ = 11.527, *p* = 0.003). For participants who identified as having advanced HIV testing knowledge, the proportion for being willing to receive ORHT, not willing and not sure was 323 (71.5%), 51 (11.3%) and 78 (17.3%) respectively, which is different from those who had low HIV testing knowledge ($\chi^2$ = 21.571, P<0.001). For participants who had advanced ORHT, the proportion for being willing to receive ORHT, not willing and not sure was 492 (68.3%), 93 (12.9%), 135 (18.8%), which is different from those who had a low knowledge of ORHT ($\chi^2$ = 28.614, P<0.001).

**Table 3. Details about HIV testing knowledge (n = 909).**

| Questions | n (answer question correctly) | % |
|---|---|---|
| AIDS can be tested by venous blood | 354 | 38.90 |
| AIDS can be tested by fingertip blood | 200 | 22.00 |
| AIDS can be tested by oral mucosa exudate | 177 | 19.50 |
| ORHT can get results quickly | 561 | 61.72 |
| ORHT doesn't need draw blood | 450 | 49.50 |
| ORHT has high accuracy | 360 | 39.60 |

**Table 4. Participants' willingness to receive an oral rapid HIV test.**

| | Willingness to receive ORHT | | | F/$\chi^2$ | P |
|---|---|---|---|---|---|
| | Yes (n$_1$ = 582) | No (n$_2$ = 129) | Unsure (n$_3$ = 198) | | |
| Age, M (SD) | 32.9(11.7) | 40.5(14.3) | 34.8(11.7) | 20.802 | <0.001 |
| Sex, n (%) | | | | 7.060 | 0.029 |
| Female | 342(66.9) | 74(14.5) | 95(18.6) | | |
| Male | 240(60.3) | 55(13.8) | 103(25.9) | | |
| Nationality, n (%) | | | | 2.278 | 0.320 |
| Han | 572(64.3) | 126(14.2) | 191(21.5) | | |
| Others | 10(50.0) | 3(15.0) | 7(35.0) | | |
| Education*, n (%) | | | | 12.381 | 0.002 |
| High school and lower | 247(58.1) | 72(16.9) | 106(24.9) | | |
| College and above | 334(69.3) | 57(11.8) | 91(18.9) | | |
| Marriage*, n (%) | | | | 19.618 | <0.001 |
| Unmarried | 258(72.3) | 32(9.0) | 67(18.8) | | |
| Married | 324(58.8) | 96(17.4) | 131(23.8) | | |
| Monthly income*, n (%) | | | | 0.897 | 0.639 |
| ≥ 5000 yuan RMB (708.4 USD) | 189(66.1) | 37(12.9) | 60(21.0) | | |
| <5000 yuan RMB (708.4 USD) | 382(63.0) | 90(14.9) | 134(22.1) | | |
| Employment Status, n (%) | | | | 4.570 | 0.102 |
| Employed | 372(66.5) | 77(13.8) | 110(19.7) | | |
| Unemployed | 210(60.0) | 52(14.9) | 88(25.1) | | |
| Type of dental setting visited, n (%) | | | | 1.052 | 0.902 |
| Dental hospital | 330(65.2) | 68(13.4) | 108(21.3) | | |
| Department of dentistry | 193(62.9) | 45(14.7) | 69(22.5) | | |
| Private dental practice | 59(61.5) | 16(16.7) | 21(21.9) | | |
| HIV Knowledge Score, M (SD) | 11.1(2.5) | 10.2(3.4) | 10.1(3.1) | 12.024 | <0.001 |
| Previously received an HIV test, n (%) | | | | 11.527 | 0.003 |
| Yes | 68(81.0) | 6 (7.1) | 10(11.9) | | |
| No | 514(62.3) | 123(14.9) | 188(22.8) | | |
| Having advanced HIV testing knowledge | | | | 21.571 | <0.001 |
| Yes | 323(71.5) | 51(11.3) | 78(17.3) | | |
| No | 259(56.7) | 78(17.1) | 120(26.3) | | |
| Having advanced ORHT knowledge | | | | 28.614 | <0.001 |
| Yes | 492(68.3) | 93(12.9) | 135(18.8) | | |
| No | 90(47.6) | 36(19.0) | 63(33.3) | | |

*Not all the participants answered the question.

All variables in Table 5 were entered in running the multiple logistic regression analysis using the forward: conditional method. Five independent variables predicted willingness to receive an ORHT (Table 6). These were: younger (OR = 0.970, 95% CI: 0.959–0.982), higher HIV knowledge score (OR = 1.087, 95% CI: 1.031–1.145), previously received an HIV test (OR = 2.057, 95% CI: 1.136–3.723), having advanced HIV testing knowledge knowing (OR = 1.570, 95% CI: 1.158–2.128), and having advanced ORHT knowledge (OR = 2.074, 95%: CI 1.469–2.928).

For the 327 participants who didn't want ORHT, we further investigated the reasons. 185 (56.6%) thought they were healthy enough and detecting HIV was unnecessary, 93 (28.4%)

**Table 5. Variables in multiple logistic regression analysis.**

| Variables | | value |
|---|---|---|
| Willingness to undergo ORHT | Y | 0 = no or unsure;1 = yes |
| Age | X1 | Actual age of participants |
| Sex | X2 | 1 = male; 2 = female |
| Nationality | X3 | 1 = han; 2 = others |
| Education | X4 | 1 = high school and low; 2 = college and above |
| Marriage | X5 | 1 = unmarried; 2 = married |
| Monthly income | X6 | 1:≤5000; 2:>5000 |
| Employment status | X7 | 0 = unemployed; 1 = employed |
| Type of dental setting visited | X9 | 1 = dental hospital; 2 = department of dentistry; 3 = private dental practice |
| HIV Knowledge Score | X10 | Actual score of HIV knowledge (range from 0 to 15) |
| Previously received an HIV test | X11 | 0 = no; 1 = yes |
| Having advanced HIV testing knowledge | X12 | 0 = no; 1 = yes |
| Having advanced ORHT knowledge | X13 | 0 = no; 1 = yes |

thought that there was no relationship between HIV/AIDS and their oral diseases, 56 (17.1%) thought it was a waste of time. Other reasons are shown in Table 7.

## Discussion

HIV testing has been a powerful measure to realize the UNAIDS 90-90-90 goal, which means 90% of all PLWH knowing their status, 90% of these individuals receiving sustained ART and 90% of those on ART having virologic suppression [38]. But in China, the diagnose rate is still far from 90% [6].

PLWH are susceptible to oral diseases, causing dental patients to become a risk group [39]. The HIV testing rate of dental patients in China was not optimistic. Li conducted a case-control study at a private dental practice in Yuxi People's Hospital in Yunnan Provence in China, 23.9% of 577 participants who refused to accept HIV screening cited they received HIV testing previously [40]. The current survey showed that 9.2% of 909 participants received HIV testing previously, and 86.9% (73/84) were tested in hospitals. HIV testing requires strong enforcement to reach the 90-90-90 goals, and besides hospitals, the dental setting can be used to screen persons for HIV.

A cross-sectional study in South Florida (U.S.) found that HIV knowledge was significantly related to previous HIV testing [41]. We found that dental patients who had higher monthly income, higher HIV/AIDS knowledge score had a higher HIV testing rate. Therefore, health

**Table 6. Predictors of willingness to receive an ORHT.**

| | B | S.E | Wals | df | Sig. | OR | 95% C.I. | |
|---|---|---|---|---|---|---|---|---|
| | | | | | | | Lower | Upper |
| Age | -0.030 | 0.006 | 25.419 | 1 | <0.001 | 0.970 | 0.959 | 0.982 |
| HIV Knowledge Score | 0.083 | 0.027 | 9.537 | 1 | 0.002 | 1.087 | 1.031 | 1.145 |
| Previously received an HIV test | 0.721 | 0.303 | 5.677 | 1 | 0.017 | 2.057 | 1.136 | 3.723 |
| Having advanced HIV testing knowledge | 0.451 | 0.155 | 8.418 | 1 | 0.004 | 1.570 | 1.158 | 2.128 |
| Having advanced ORHT knowledge | 0.730 | 0.176 | 17.205 | 1 | <0.001 | 2.074 | 1.469 | 2.928 |
| Constant | -0.085 | 0.367 | 0.053 | 1 | 0.817 | 0.919 | | |

**Table 7. Major reasons participated cited for not wanting a ORHT (n = 327).**

| Reasons | n | % |
|---|---|---|
| Healthy enough | 185 | 56.6 |
| Not related to their dental diseases | 93 | 28.4 |
| Waste of time | 56 | 17.1 |
| Worried about the inaccuracy of the test | 52 | 15.9 |
| Not able to bear the result (if reactive) | 26 | 8.0 |
| Worry about the discrimination | 26 | 8.0 |
| Others | 20 | 6.1 |

education among dental patients can improve their awareness of HIV/AIDS and further increase their testing rate.

As ORHT is easy to administer and results can be read within 20 minutes, several researchers had explored the ORHT acceptance [12, 42, 43]. A study conducted in a hospital emergency department in Boston (U.S.) revealed that 68.8% of 821 responders are willing to accept ORHT, similar to that of finger-stick testing [43]. For most-at-risk people, two cross-sectional studies conducted in Beijing (China) and Shandong (China) suggested that the ORHT acceptance rate was 85.1% (223/262) and 70.9% (806/1137), respectively [12, 42]. Our study found that 64.0% of 909 participants were willing to receive an ORHT.

In the current study, we observed that those with high HIV knowledge scores, having advanced HIV testing and ORHT knowledge, experience of HIV testing had more willingness than those who hesitated or rejected to receive ORHT. Improving HIV-related knowledge is an essential and inexpensive way that can influence the acceptance rate. In addition to HIV knowledge, HIV testing knowledge and ORHT knowledge cannot be ignored, it should also be an important content of health education. Participants may change their mind if they recognize ORHT as a rapid, accurate and moderately priced HIV screening method. According to select U.S. studies, 84.6% (225/266) participants reported they would likely accept the HIV test if they received a physician's recommendation [44]. Studies have shown that dentist were willing to carry out ORHT among their patients [29]. If recommended by a dentist, dental patients' willingness to receive ORHT will likely improve.

Barriers for ORHT acceptance among dental patients included being tested for HIV previously, feeling confident about their health status, lack of time to wait for the test result, not wanting to be tested in a private dental practice and expensive testing cost [34, 40]. In our study, 56.6% of 327 participants believed they were sufficiently healthy indicating they were not interested in receiving an ORHT. This is inconsistent with the body of literature that has established the role of discrimination in persons rejecting HIV test offers [45]. One reason for not citing discrimination has a barrier to receiving an ORHT may be because the participants were established patients of the clinic where they were surveyed and satisfied with the quality of dental services they were receiving from their providers. Over a quarter (28.4%) of 327 participants thought that ORHT is not related with their oral diseases, from which we can speculated that some participants were not familiar with the oral manifestations of HIV. In fact, private dental practice are not risk-free places for HIV. Several participants expressed concern about the ORHT inaccuracy and not being a good use of their time. Therefore, carrying out health education on HIV and ORHT is necessary to improve the sense of risk of HIV and knowledge degree of ORHT to improve the acceptance.

For most-at-risk populations (including men who have sex with men (MSM), female sex workers and voluntary counseling and testing clients) in Shandong, a province in eastern China, the median price they would pay for ORHT ranged from 34 to 57 yuan RMB (4.8 to 8.2

USD) [42]. Moreover, the median price for MSM in Beijing (China) who were willing to pay was 57 yuan RMB (8.1 USD) [12]. In current study, 46.1% (18/39) participants who tested for HIV previously paid for more than 51 yuan RMB (7.3 USD). And for the 582 participants who were willing to receive an ORHT, the median price they were willing to pay was 30 yuan RMB (4.3 USD), lower than the mentioned studies among most-at-risk people. More specifically, 69.8% of the 580 participants accepting ORHT in the dental setting would accept HIV testing at a cost of 30 yuan RMB (4.3 USD), and 49.3% would be willing to pay 50 yuan RMB (7.2 USD) for ORHT. Undoubtedly, more acceptance would be gained if the price become lower.

Additional studies are needed to implement ORHT in dental practices and assess HIV prevalence rate among dental patients, and evaluating the prospect of promoting ORHT in the dental setting. In the future, our study team plans to select dental venues in high prevalent neighborhoods to assess the feasibility of implementing ORHT among dental patients.

Study limitations include (1) most of the dental patient participants were from local jurisdictions and their results may not be generalizable to the larger population, (2) we recruited less participants from private dental practices because some private dental practice refused to participate because of fear of affecting the patient's medical experience, which may lead to some section bias, and (3) lack of information on survey non-responders.

## Conclusion

The majority of dental patients had not previously received an HIV test, although many were receptive to being tested in the dental setting. The dental setting as a venue to screen people for HIV needs further exploration, particularly because many people do not associate dentistry with HIV and other chairside screenings for chronic diseases. Increasing awareness of ORHT and reducing testing price can further improve the patient's willingness to receive ORHT.

## Supporting information

**S1 Data.**
(SAV)

**S1 Questionnaire.**
(DOCX)

**S2 Questionnaire.**
(DOCX)

## Author Contributions

**Conceptualization:** Lirong Wang, Anthony J. Santella, Guihua Zhuang, Ruizhe Huang.

**Data curation:** Bei Gao, Lirong Wang.

**Formal analysis:** Bei Gao.

**Investigation:** Bei Gao, Boya Xu, Yujiao Liu, Shuya Xiao, Shifan Wang.

**Methodology:** Lirong Wang.

**Project administration:** Lirong Wang.

**Writing – original draft:** Bei Gao.

**Writing – review & editing:** Lirong Wang, Anthony J. Santella.

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
