## [Decision Letter · Decision Letter 0]

16 Nov 2020

PONE-D-20-30084

HIV Testing Behaviors and Willingness to Receive Oral Rapid HIV Testing among Dental Patients in Xi’an, China

PLOS ONE

Dear Dr. Wang,

Thank you for submitting your manuscript to PLOS ONE. After careful consideration, we feel that it has merit but does not fully meet PLOS ONE’s publication criteria as it currently stands. Therefore, we invite you to submit a revised version of the manuscript that addresses the points raised during the review process.

We look forward to receiving your revised manuscript.

Kind regards,

Zixin Wang, PhD.

Academic Editor

PLOS ONE

Additional Editor Comments:

1. Given the very low overall HIV prevalence in China, more justification is needed to support the use of dental clinics settings to increase HIV testing rate. If the aim is to increase HIV testing rate at population level, offering HIV testing at primary healthcare settings would be far more efficient.

2. Opt-out HIV testing is commonly used at clinical settings in some western countries to increase testing rate, which means every patients will receive HIV testing unless they show objection. It is unclear whether opt-in HIV testing (like the author described) would be useful to increase HIV testing, as those who have sexual risk behaviors may choose not to accept the tests. The authors should explain why they look at willingness to use opt-in HIV testing, instead of the more commonly used opt-out HIV testing in clinical settings.

3. The inclusion criteria is a bit confusing. The authors mentioned all dental patients were invited. It is unclear whether the study recruit those under 18 years old?

4. Details of the introduction about ORHT should be included in the manuscript.

5. I read through the manuscript and I was not able to find out how willingness to use ORHT is measured. I do agree with the reviewer that a detailed description of measurements should be included in the manuscript, instead of only referring to some published papers or just mentioning number of items.

Journal Requirements:

2. You have mentioned that the questionnaire was validated, but not whether it was  pre-tested. Please clarify if it was pre-tested. If this did not occur, please provide the rationale for not doing so.

3. Please correct your reference to "p=0.000" to "p<0.001" or as similarly appropriate, as p values cannot equal zero.

4. Please provide further details on sample size and power calculations.

Reviewers' comments:

Reviewer's Responses to Questions

**Comments to the Author**

1. Is the manuscript technically sound, and do the data support the conclusions?

Reviewer #1: Yes

2. Has the statistical analysis been performed appropriately and rigorously? 

Reviewer #1: Yes

3. Have the authors made all data underlying the findings in their manuscript fully available?

Reviewer #1: Yes

4. Is the manuscript presented in an intelligible fashion and written in standard English?

Reviewer #1: Yes

5. Review Comments to the Author

Reviewer #1: This manuscript has a clear and straightforward research question. It aims to assess the HIV testing behaviors and willingness to receive ORHT among dental patients in Xi’an, China.

However, the following issues should be addressed.

1.Measurements should be described in more detailed in the Method section. For instance, were the items rated based on Likert scale? Also, please give examples of the items in the Method section, especially the items about ORHT knowledge and willingness to receive ORHT.

2. What were the reliabilities of the measurements?

3.Can the authors think about presenting a table about the results of each item in percentage. For instance, Q37, what is the percentage of participants chose option 1, 2 and 3 respectively? By having such information, the readers can have a better understanding of the study results.

4.Only 8% of those who were unwilling to receive ORHT chose worrying about being discriminated. In the literature, as fear of being discriminated is one of the major factors for not willing to do HIV test. As 8% in study is relatively low, could the authors give the reasons for this phenomenon?

5.Apart from the variables that were presented in Table 4, any other variables were entered in doing the multiple logistic regression analysis? If so, what are they? Were all the variables that were significant in chi-squared test entered in the multiple logistic regression analysis? Can the authors give a brief description of the rationale for inclusion of variables in the regression analysis?

6.Only 10.6% of the participants were from private dental practice. How this figure would affect the result? Is there any statistics about the number of people attending different types of dental clinics in the study place, Xi’an? If there is a higher proportion of people attending private dental clinics than other types, the results of this study may be distorted. Thus, such kind of information is important.

6. PLOS authors have the option to publish the peer review history of their article (what does this mean?). If published, this will include your full peer review and any attached files.

Reviewer #1: No

---

## [Author Response · Author response to Decision Letter 0]

30 Jan 2021

Response to reviewers' comments:

Comment 1: Given the very low overall HIV prevalence in China, more justification is needed to support the use of dental clinics settings to increase HIV testing rate. If the aim is to increase HIV testing rate at population level, offering HIV testing at primary healthcare settings would be far more efficient.

Response to reviewer:

Many thanks for your constructive comment. We have added the relevant information to the Introduction section: “The Chinese government has explored the feasibility of offering HIV testing at primary healthcare settings such as community health centers [11]. However, since China is a low-prevalence country, such a practice is still being explored. More research is needed to evaluate the effectiveness of this setting and other non-traditional settings such as dentistry.” （Page 4 Line 75-79）

[11]. Ong JJ, Peng MH, Wong WW, Lo Y-R, Kidd MR, Roland M, et al. Opportunities and barriers for providing HIV testing through community health centers in mainland China: a nationwide cross-sectional survey. BMC Infectious Diseases. 2019;19(1):1054. doi: 10.1186/s12879-019-4673-0.

Comment 2: Opt-out HIV testing is commonly used at clinical settings in some western countries to increase testing rate, which means every patient will receive HIV testing unless they show objection. It is unclear whether opt-in HIV testing (like the author described) would be useful to increase HIV testing, as those who have sexual risk behaviors may choose not to accept the tests. The authors should explain why they look at willingness to use opt-in HIV testing, instead of the more commonly used opt-out HIV testing in clinical settings.

Response to reviewer:

We thank the reviewers for their comments and suggestions. We have added the relevant information to the Introduction section: “To expand testing venues, opt-out HIV testing has been implemented in medical settings among priority populations since 2008 and settings such as perinatal exams, prenatal exams, pre-operative exams, sexually transmitted disease exams and other types of testing services in China [9]. Almost half (44.7%) of 528,234 persons diagnosed with HIV between 2006 and 2014 in China were diagnosed at hospitals, indicating that most of the HIV testing in China was conducted in health and hospital systems [10]. However, dental patients are nor high-risk groups or sentinel surveillance groups. For both opt-out and opt-in testing, it is necessary to first understand the population’s HIV testing willingness. Otherwise, even if opt-out is carried out, a considerable number of people may refuse to be tested.” （Page 3 Line 66-75）

[9]. Jin Xia, Xiong Ran, Yurong M. HIV/AIDS cases detection in medical institutions from 2008 to 2013 in China. Chinese Journal of Epidemiology. 2015;36(4):323-6.

[10]. Tang H, Mao Y, Tang W, Han J, Xu J, Li J. Late for testing, early for antiretroviral therapy, less likely to die: results from a large HIV cohort study in China, 2006-2014. BMC infectious diseases. 2018;18(1):272. doi: 10.1186/s12879-018-3158-x. PubMed PMID: 29895275.

Comment 3: The inclusion criteria is a bit confusing. The authors mentioned all dental patients were invited. It is unclear whether the study recruit those under 18 years old?

Response to reviewer:

Many thanks for your valuable suggestions. To clarify, we did not invite those under 18 years old to participant our survey. We have revised the Methods section: “Dental patients 18 years and older who visited the sample practices from August to September were invited to participate in the survey”. （Page 6 Line 122-123）

Comment 4: Details of the introduction about ORHT should be included in the manuscript.

Response to reviewer:

Thank you for the suggestions pointed by the reviewer. We have added the relevant information in the Introduction section: “The first oral rapid HIV test (ORHT) was approved by the United States (US) Food and Drug Administration (FDA) in 2004[17]. In the US and other high-income countries, scholars have proposed that ORHT has the potential to improve the diagnosis rate of PLWH [18, 19]. The tests have high sensitivity and specificity; both above 99%[20-22]. It also offers many advantages over blood-based testing and could address several challenges in implementing HIV testing in settings without laboratories [20]. This testing method does not require blood collection, has no risk of occupational exposure and cross-infection, and can prevent the patient from refusing to test because of fear of blood draw [23].” （Page 4 Line 85-92） 

[17]. Food Drug Administration Based FAFOFJB, Maryland: US Department of Health Human Services. Rapid HIV Test Kit. 2004.

[18]. Parish CL, Siegel K, Liguori T, Abel SN, Pollack HA, Pereyra MR, et al. HIV testing in the dental setting: perspectives and practices of experienced dental professionals. AIDS Care. 2017;30(3):347-52. doi: 10.1080/09540121.2017.1367087.

[19]. Chung R, Leung SJ, Abel SN, Hatton MN, Ren Y, Seiver J, et al. HIV screening in the dental setting in New York State. PloS one. 2020;15(4):e0231638. doi: 10.1371/journal.pone.0231638. PubMed PMID: 32298336.

[20]. Belete W, Deressa T, Feleke A, Menna T, Moshago T, Abdella S, et al. Evaluation of diagnostic performance of non-invasive HIV self-testing kit using oral fluid in Addis Ababa, Ethiopia: A facility-based cross-sectional study. PloS one. 2019;14(1):e0210866. doi: 10.1371/journal.pone.0210866. PubMed PMID: 30682062.

[21]. Delaney K, Branson B, Uniyal A, Phillips S, Candal D, Owen S, et al. Evaluation of the performance characteristics of 6 rapid HIV antibody tests. Clinical infectious diseases : an official publication of the Infectious Diseases Society of America. 2011;52(2):257-63. doi: 10.1093/cid/ciq068. PubMed PMID: 21288853.

[22]. Delaney KP, Branson BM, Uniyal A, Kerndt PR, Keenan PA, Jafa K, et al. Performance of an oral fluid rapid HIV-1/2 test: experience from four CDC studies. Aids. 2006;20(12):1655-60. Epub 2006/07/27. doi: 10.1097/01.aids.0000238412.75324.82. PubMed PMID: 16868447.

[23]. Zachary D, Mwenge L, Muyoyeta M, Shanaube K, Schaap A, Bond V, et al. Field comparison of OraQuick ADVANCE Rapid HIV-1/2 antibody test and two blood-based rapid HIV antibody tests in Zambia. BMC infectious diseases. 2012;12(8):183. doi: 10.1186/1471-2334-12-183. PubMed PMID: 22871032. 

Comment 5: I read through the manuscript and I was not able to find out how willingness to use ORHT is measured. I do agree with the reviewer that a detailed description of measurements should be included in the manuscript, instead of only referring to some published papers or just mentioning number of items.

Response to reviewer:

We agree with that a detailed description of measurements should be included in the manuscript, so we revised the Methods section as follows: The survey instrument was modified from a validated questionnaire from researchers in the U.S. [33]. An expert consensus panel of Chinese dental, sexual health, and epidemiology scholars reviewed a draft questionnaire to conform the validity. The questionnaire was pre-tested among 20 patients in Xi’an, and we modified the questionnaire based on the results, including the order of the questions, some inappropriate options and inaccurate expression of the questions. The final questionnaire consisted of 44 questions, assessing socio-demographic characteristics (eight items), HIV/AIDS knowledge (modified HIV-KQ-18 [37] (fifteen items), HIV testing behaviors (seven items), HIV testing knowledge (six items), ORHT knowledge (three items) and the willingness to receive ORHT (five items). We assessed ORHT knowledge by asking participants what the advantages of ORHT were, the options were high accuracy, quick results, and no need to draw blood. To collect the willingness to receive ORHT, we asked the participants: “If a dentist can carry out Oral Rapid HIV Testing, would you be willing to test it before treating oral diseases?”. The answers were yes, unwilling and unsure. For participants who chose “yes”, we categorized them as willing to use ORHT. （Page 7 Line 139-152） 

[33]. Davide SH, Santella A J, Furnari W, Leuwaisee P, Cortell M, Krishnamachari B. Patients' Willingness to Participate in Rapid HIV Testing: A pilot study in three New York City dental hygiene clinics. Journal of dental hygiene. 2017;91(6):41-8. PubMed PMID: 29378805.

[37]. Carey M, Schroder K. Development and psychometric evaluation of the brief HIV Knowledge Questionnaire. AIDS education. 2002;14(2):172-82. doi: 10.1521/aeap.14.2.172.23902. PubMed PMID: 12000234. 

Comment 6: You have mentioned that the questionnaire was validated, but not whether it was pre-tested. Please clarify if it was pre-tested. If this did not occur, please provide the rationale for not doing so.

Response to reviewer:

Thank you for pointing out this issue. We pre-tested the questionnaire before the survey. We revised the Method section as follows: “The questionnaire was pre-tested among 20 patients in Xi’an, and we modified the questionnaire based on the results, including the order of the questions, some inappropriate options and inaccurate expression of the questions.” (Page 7 Line 141-144) 

Comment 7: Please correct your reference to "p=0.000" to "p<0.001" or as similarly appropriate, as p values cannot equal zero.

Response to reviewer:

Many thanks for pointing out errors. We have revised all “p=0.000” to “p<0.001”.

Comment 8: Please provide further details on sample size and power calculations.

Response to reviewer:

We added the sample size details into the Methods section: “The proportion of participants who were willing to receive ORHT ranged from 24% to 91% [36]. If we take 24% as the expected proportion of our survey, α= 0.01 and the allowable error d=0.15P, according to the formula , which is commonly used to estimate the sample size of cross sectional study, then the approximately sample size would be 934.” （Page 6 Line 123-127） 

[36]. Christopoulos K, Weiser S, Koester K, Myers J, White D, Kaplan B, et al. Understanding patient acceptance and refusal of HIV testing in the emergency department. BMC public health. 2012;12(3):1471-2458. doi: 10.1186/1471-2458-12-3. PubMed PMID: 22214543. 

Comment 9: Measurements should be described in more detailed in the Method section. For instance, were the items rated based on Likert scale? Also, please give examples of the items in the Method section, especially the items about ORHT knowledge and willingness to receive.

Response to reviewer:

The items of our survey did not base on Likert scale, but we used three categories of responses (described above). Since we focused on the willingness of dental patients, we intended to ask about the specific acceptance level after the willingness, such as cost. We added details as to how we measure ORHT knowledge and willingness to receive ORHT in the method sections (see response above). We feel the revised Methods section addresses this concern.

Comment 10: What were the reliabilities of the measurements?

Response to reviewer:

As you know, reliability refers to the extent to which a scale produces consistent results, if the measurements are repeated a number of times. There are four different approaches: Test-Retest, Internal Consistency Reliability, Split Half Reliability and Inter-Rater Reliability. We considered reliability during the pre-test. However, dental patients are of strong fluidity. It would be virtually impossible and logistically burdensome for us to survey dental patients repeatedly to evaluate the reliability. This is a limitation of the current study.

Comment 11: Can the authors think about presenting a table about the results of each item in percentage. For instance, Q37, what is the percentage of participants chose option 1, 2 and 3 respectively? By having such information, the readers can have a better understanding of the study results.

Response to reviewer:

Thank you very much for pointing out this issue. We added the percentage of each item in the result part: “In details, for the advantages of ORHT, the number of choosing “quick results”, “no need to draw blood” and “high accuracy” are 561(61.7%) ,450(49.5%) and 360(39.6%), respectively. And other questions are showed in Table 3.” (Page 11 Line 202-205).

Table 3 Details about HIV testing knowledge (N=909)

Questions n (answer question correctly) %

AIDS can be tested by venous blood 354 38.90

AIDS can be tested by fingertip blood 200 22.00

AIDS can be tested by oral mucosa exudate 177 19.50

ORHT can get results quickly 561 61.72

ORHT doesn’t need draw blood 450 49.50

ORHT has high accuracy 360 39.60

Comment 12: Only 8% of those who were unwilling to receive ORHT chose worrying about being discriminated. In the literature, as fear of being discriminated is one of the major factors for not willing to do HIV test. As 8% in study is relatively low, could the authors give the reasons for this phenomenon?

Response to reviewer:

Thanks very much for your important question. We have modified the relevant information in the Discussion section: “In our study, 56.6% of 327 participants believed they were sufficiently healthy indicating they were not interested in receiving an ORHT. This is inconsistent with the body of literature that has established the role of discrimination in persons rejecting HIV test offers [45]. One reason for not citing discrimination has a barrier to receiving an ORHT may be because the participants were established patients of the clinic where they were surveyed and satisfied with the quality of dental services they were receiving from their providers.” (Page 18 Line 287-293).

[45]. Batey D, Hogan V, Cantor R, Hamlin C, Ross-Davis K, Nevin C, et al. Short communication routine HIV testing in the emergency department: assessment of patient perceptions. AIDS research human retroviruses. 2012;28(4):352-6. doi: 10.1089/aid.2011.0074. PubMed PMID: 21790474. 

Comment 13: Apart from the variables that were presented in Table 4, any other variables were entered in doing the multiple logistic regression analysis? If so, what are they? Were all the variables that were significant in chi-squared test entered in the multiple logistic regression analysis? Can the authors give a brief description of the rationale for inclusion of variables in the regression analysis?

Response to reviewer:

We added the following sentence to Results section to provide clarification: “All variables in Table 5 were entered in running the multiple logistic regression analysis using the forward: conditional method.” (Page 15 Line 233-234).

Table 5 Variables in multiple logistic regression analysis

Variables value

Willingness to undergo ORHT Y 0=no or unsure;1=yes

Age X1 Actual age of participants 

Sex X2 1=male; 2=female

Nationality X3 1=han; 2=others

Education X4 1=high school and low; 2=college and above

Marriage X5 1=unmarried; 2=married

Monthly income X6 1=no more than 5000; 2=less than 5000

Employment status X7 0=unemployed; 1=employed

Type of dental setting visited X9 1=dental hospital; 2=department of dentistry; 3=private dental practice

HIV Knowledge Score X10 Actual score of HIV knowledge (range from 0 to 15)

Previously received an HIV test X11 0=no; 1=yes

Having advanced HIV testing knowledge X12 0=no; 1=yes

Having advanced ORHT knowledge X13 0=no; 1=yes

 

Comment 14: Only 10.6% of the participants were from private dental practice. How this figure would affect the result? Is there any statistics about the number of people attending different types of dental clinics in the study place, Xi’an? If there is a higher proportion of people attending private dental clinics than other types, the results of this study may be distorted. Thus, such kind of information is important.

Response to reviewer:

Private dental practices in Xi’an are dispersive and have an irregular flow of patients. Additionally, dental hospitals and departments of dentistry in general hospitals are usually at capacity and represent the major source for dental care. This was the main reason that only 10.6% of the participants were from private dental practice. However, fifty-nine (61.46%) of the 96 participants from private dental practice were willing to receive ORHT before dental therapy, similar to the percentage in other types of dental settings. Based on the results of our data, the percentage of participants from private dental practice have little influence on the willingness of the whole dental patients. According to our design, the sequence of promoting ORHT among dental patients is from dental hospitals, departments of dentistry in general hospital to private dental practice. The results of this research are essential for the further promotion of ORHT among dental patients. In addition, we added the following to address this concern in the Discussion section: “We recruited less participants from private dental practices because some private dental practice refused to participate because of fear of affecting the patient's medical experience, which may lead to some section bias.” (Page 20 Line 316-319).

---

## [Decision Letter · Decision Letter 1]

2 Mar 2021

HIV Testing Behaviors and Willingness to Receive Oral Rapid HIV Testing among Dental Patients in Xi’an, China

PONE-D-20-30084R1

Dear Dr. Wang,

We’re pleased to inform you that your manuscript has been judged scientifically suitable for publication and will be formally accepted for publication once it meets all outstanding technical requirements.

Kind regards,

Zixin Wang, PhD.

Academic Editor

PLOS ONE

Additional Editor Comments (optional):

Reviewers' comments:

Reviewer's Responses to Questions

**Comments to the Author**

1. If the authors have adequately addressed your comments raised in a previous round of review and you feel that this manuscript is now acceptable for publication, you may indicate that here to bypass the “Comments to the Author” section, enter your conflict of interest statement in the “Confidential to Editor” section, and submit your "Accept" recommendation.

Reviewer #1: All comments have been addressed

2. Is the manuscript technically sound, and do the data support the conclusions?

Reviewer #1: Yes

3. Has the statistical analysis been performed appropriately and rigorously? 

Reviewer #1: Yes

4. Have the authors made all data underlying the findings in their manuscript fully available?

Reviewer #1: Yes

5. Is the manuscript presented in an intelligible fashion and written in standard English?

Reviewer #1: Yes

6. Review Comments to the Author

Reviewer #1: (No Response)

7. PLOS authors have the option to publish the peer review history of their article (what does this mean?). If published, this will include your full peer review and any attached files.

Reviewer #1: No

---

## [Editor Report · Acceptance letter]

16 Mar 2021

PONE-D-20-30084R1 

HIV Testing Behaviors and Willingness to Receive Oral Rapid HIV Testing among Dental Patients in Xi’an, China 

Dear Dr. Wang:

I'm pleased to inform you that your manuscript has been deemed suitable for publication in PLOS ONE. Congratulations! Your manuscript is now with our production department. 

Kind regards, 

on behalf of

Professor Zixin Wang 

Academic Editor

PLOS ONE